# Dynamics of Physicochemical Properties, Functional Compounds and Antioxidant Capacity during Spontaneous Fermentation of *Lycium ruthenicum* Murr. (Qinghai–Tibet Plateau) Natural Vinegar

**DOI:** 10.3390/foods11091344

**Published:** 2022-05-05

**Authors:** Qingchao Gao, Yangbo Song, Ying Liang, Yahui Li, Yingjiu Chang, Rong Ma, Xiaohai Cao, Shulin Wang

**Affiliations:** 1College of Agriculture and Animal Husbandry, Qinghai University, Xining 810016, China; gqc199411@163.com (Q.G.); yangbosong792@163.com (Y.S.); 18797334618@163.com (Y.C.); 13649716346@163.com (R.M.); 15709781687@163.com (X.C.); 2Key Laboratory of Food Quality and Safety of Jiangsu Province, Institute of Food Safety and Nutrition, Jiangsu Academy of Agricultural Science, Nanjing 210014, China; lyjaas@163.com (Y.L.); liqianhao217@126.com (Y.L.); 3United Laboratory of Food Research and Testing of Qinghai-Gansu Province, Xining 810016, China; 4State Key Laboratory of Plateau Ecology and Agriculture, Qinghai University, Xining 810016, China

**Keywords:** *Lycium ruthenicum* Murr., fermentation profile, metabolite, antioxidant capacity, correlation

## Abstract

Functional fermented fruit drinks are known worldwide for their health-promoting potential. Black wolfberry (BW) has high nutritional value, and its relative product development can be enriched through fermentation technology, so that its market might be broadened. Total acid, sugars, proteins, enzymes, anthocyanins, flavonoids, polyphenols, organic acids and DPPH free radical scavenging ability (DPPH) were tracked and determined by colorimetric method and HPLC during spontaneous fermentation of BW vinegar. The antioxidant capacity in vitro of BW vinegar was evaluated based on the dynamics of antioxidant contents and DPPH. The results showed that total acid continuously increased during fermentation, yet total sugar and reducing sugar shared a similar decreasing trend. The composition of samples differed in terms of total anthocyanins, total flavonoid, total polyphenol, total protein, superoxide dismutase (SOD), amylase, organic acids and DPPH through spontaneous fermentation. Functional compounds including total polyphenol, total flavonoid and three organic acids (γ-aminobutyric acid, lactic acid and gallic acid) played the main roles in antioxidation. Unexpectedly, SOD and ascorbic acid as antioxidants did not correlate with DPPH, but they were rich in the final products at 754.35 U/mL and 3.39 mg/mL, respectively. Generally, the quality of BW vinegar has been improved based on analyzing dynamics on functional compounds, organic acids and antioxidant capacity, which proves that BW vinegar obtained by spontaneous fermentation should be a potential source of fermented food with antioxidant effects for consumers.

## 1. Introduction

Black wolfberry (BW, *Lycium ruthenicum* Murr.) is a nutritional and dark purple berry of Solanaceae, widely distributing in the Qinghai–Tibet Plateau. BW is a traditional medicinal and food plant since it is rich in nutritional and functional components [1,2,3]. Although the beneficial effects presented by BW are noticeable, there is a scarcity of by-products made from BW.

Vinegar fermentation may occur spontaneously and be induced by back-slopping or starters, in both small- and large-scale production. As one of the fermented products, natural vinegar has some potential from a nutraceutical standpoint. Historically, natural vinegar was discovered in agriculture, along with the production of alcoholic fermentation of grains, fruits and vegetables [4]. In recent years, natural vinegars have been produced from fruit sources such as grape, lemon [5], kiwifruit, apple, persimmon [6] and pomegranate [7], as well as from other untypical mini fruits, such as fig, mulberry [8], plum [9], raspberry [10], gooseberry [11], cherry [12] and blueberry [13]. Natural vinegar contains biologically active ingredients, for instance, polyphenol, anthocyanin, flavonoid compounds and organic acids, which show strong antioxidant activity [14], antimicrobial activity [12], antihypertensive effect [15], antitumor activity [16] and blood glucose control [17]. Supplementation with natural dietary antioxidants has become an essential strategy in the treatment of diet-related diseases, reducing oxidative stress, preventing age-related diseases and maintaining homeostasis [16]. Hence, natural vinegar has not only been used as a condiment and food preservative for thousands of years [18], but it is also used in traditional folk medicine [19] and beauty products [20] in modern times. Nowadays, the consumption of natural vinegar has been increased due to its health benefits.

Among naturally fermented vinegar, Komesu and Kurosu (rice vinegar in Japan) are known with respect to their health benefits through preventing inflammation and hypertension. Kibizu (sugar cane vinegar in Amami Ohshima, Japan), Kurosu (black rice vinegar in Kagoshima, Japan), Kouzu (black rice vinegar in China) and red wine vinegar in Italy have potent radical-scavenging activity as analyzed by the DPPH method [4]. Spontaneous fermentation is suitable for small-scale production and only for very specific juices. The quality of natural vinegar is determined by raw materials, fermentation methods and aging [6]. Positive health effects of natural berry vinegar consumption could be related with the conversion process during whole fermentation that leads to an augment in bioactive compounds contents and a change in the composition of polyphenols, anthocyanins and flavonoid compounds, SOD and organic acids [21].

Earlier research on BW has focused on the detection and identification of functional components (anthocyanins, polyphenols and organic acids) and physicochemical properties in raw materials [16,22,23] and products based on pretreatment and processing [24,25]. To the best of our knowledge, no research on the interaction between bioactive components and characteristic physicochemical properties of natural vinegar made from BW during spontaneous fermentation has been reported. Thus, this current study aims to evaluate the dynamics of physicochemical properties (total acid, total sugar, reducing sugar and total protein), functional compounds (total anthocyanins, total flavonoid, total polyphenol, SOD, amylase and organic acids) and potential health effects (antioxidant capacity). Simultaneously, the effect of interaction between metabolites and antioxidant capacity during spontaneous fermentation of BW vinegar was investigated. This work will provide a theoretical reference for the production and characterization of BW products from the Qinghai–Tibet Plateau in the future.

## 2. Materials and Methods

### 2.1. Chemicals and Reagent

The kits for estimating SOD, amylase kit and coomassie brilliant blue kit were purchased from Nanjing Jiancheng Bioengineering Institute (Nanjing, China). Rutin (153-18-4) and gallic acid (149-91-7) standards were purchased from Chengdu DeSiTe Biological Technology Co., Ltd. (Chengdu, China). Glacial acetic acid standard (64-19-7) was purchased from Shanghai Acmec Biochemical Co., Ltd. (Shanghai, China). Twelve kinds of organic acid standards, namely, lactic acid (50-21-5), gallic acid (149-91-7), fumaric acid (110-17-8), sorbic acid (110-44-1), ascorbic acid (50-81-7), succinic acid (110-15-6), oxalic acid (144-62-7), shikimic acid (138-59-0), L-malic acid (97-67-6), citric acid (77-92-9), L-tartaric acid (87-69-4) and γ-aminobutyric acid (56-12-2) and foline-phenol were purchased from Beijing Solabao Science and Technology Co., Ltd. (Beijing, China). DPPH free radical was purchased from Tokyo Chemical Industry (Tokyo, Japan). All other chemicals and reagents used in this study were of analytical grade.

### 2.2. BW Vinegar Fermentation

Fresh BW was obtained from a farm (36°17′ N, 95°46′ E, Altitude 3556 m), Golmud region, Qinghai, China, on 23 August 2018. All berries without disease were ripe and of excellent quality. The whole berry with skins and seeds was used to prepare the vinegar (the stalks were removed). The vinegar fermentation process was performed using the description of Ozturk et al. [5] with some modification. Crushed berries and sterile water were used at a ratio of 0.425:1 (*w*/*w*). Vinegar was produced via spontaneous fermentation performed in glass jars at 28 ℃ for 60 days, carried out by the natural flora inhabiting the berry in an ultra-clean bench, and the jars were sealed and placed in a dark temperature-controlled incubator (SPX-450FT, Ningbo Prand Instrument Co., Ltd., Zhejiang, China). The fermentation process was performed in triplicate, and 3 samples (one sample, 50 mL) in each fermentation process were collected at 0, 5th, 10th, 15th, 20th, 25th, 30th, 35th, 40th, 45th, 50th, 55th and 60th days, respectively. The collected samples were filtered with 4 layers of skimmed gauze and centrifuged at 3000 r/min for 10 min. The supernatants were collected and stored at −80 °C until use. All the sampling procedures were carried out in the clean bench.

### 2.3. Measurement of Fermentation Parameters

To determine total acid, 4 g sample was titrated against 0.1 M NaOH to a final pH of 8.5. The amount of consumed NaOH (mL) was recorded, and total acid content was calculated based on the conversion coefficient of lactic acid, expressed as mg/mL. Total sugar content was measured by the anthrone-sulphuric acid method [26]. Reducing sugar was determined using the 3,5-dinitrosalicylic acid (DNS) method [27]. Using glucose as a standard, standard curves of total sugar and reducing sugar were obtained, *y* = 7.3567*x* − 0.012, (mg/mL, *R*^2^ = 0.9906) and *y* = 2.1593*x* − 0.0074, (mg/mL, *R*^2^ = 0.9958), respectively.

### 2.4. Measurement of Total Anthocyanins Content

Total anthocyanins content was measured by the pH differential method described at the literature [28]. Briefly, 0.1 mL sample was diluted into 10 mL of aqueous buffer (pH 1.0) and sodium acetate buffer (pH 4.5), respectively. After 50 min equilibration, they were determined at 535 nm and 700 nm, respectively. The formula 3 for calculating total anthocyanins is as follows, where *A* is (Absorbance_535nm_–Absorbance_700nm_) pH 1.0—(Absorbance_535nm_–Absorbance_700nm_) pH 4.5:(1)$$TA=\frac{A\times M_{W}\times DF}{\varepsilon \times L\times V}\;$$

To measure total anthocyanins (*TA*) as Cyanidin-3-glucoside equivalent (CGE), the amount was calculated using the molar absorptivity (*ε* = 26,900) (L/(moL·cm)) and molecular weight (*M_W_* = 449.2) of CGE. *L* is the light path (1 cm), *DF* is dilute-fold (100) and *V* is sample volume (mL). Total anthocyanins content was expressed as mg CGE/mL.

### 2.5. Measurement of Total Flavonoid Content

The sodium nitrite-aluminum and nitrate-sodium hydroxide colorimetry method was used to determine total flavonoid content of the samples [29] with minor modification. Briefly, each sample (2 mL) was mixed with 0.3 mL 5% NaNO_2_ in each tube, and they were left to stand still for 5 min. Subsequently, 0.3 mL 10% Al(NO_3_)_3_ was added with a standing of 6 min, 4.0 mL 4% NaOH for again 10 min. Finally, the absorbance values were measured at 510 nm. Using rutin as a standard, total flavonoid content was calculated from the calibration curve *y* (absorbance) = 11.403*x* (rutin equivalents content) + 0.0182 (*R*^2^ = 0.9968), expressed as rutin mg equivalents per 1 mL.

### 2.6. Measurement of Total Polyphenol Content

Total polyphenol content was measured referring to the Folin–Ciocalteu method [30]. Briefly, 4.5 mL Folin–Ciocalteu reactive (FCR) was added into a 0.5 mL sample. After 5 min, 4 mL Na_2_CO_3_ (7.5%) was added. Subsequently, the mixture was shaken on a shaker and let to stand still for 1 h in dark at room temperature, then absorbance was measured at 760 nm. Gallic acid (GA) was used as a standard compound. Total polyphenol content was calculated from the calibration curve *y* (absorbance) = 0.1014*x* (GA equivalents content) + 0.0444 (*R*^2^ = 0.9994), expressed as GA mg equivalents per 1 mL.

### 2.7. Measurement of Total Protein, SOD and Amylase

Determinations of total protein, SOD and amylase content were performed with the measurement kits supplied by Nanjing Jiancheng Bioengineering Institute (Nanjing, China).

### 2.8. Measurement of Organic Acids

Thirteen organic acids, including γ-aminobutyric acid (GABA), tartaric acid, citric acid, malic acid, shikimic acid, lactic acid, acetic acid, oxalic acid, succinic acid, ascorbic acid, sorbic acid, fumaric acid and gallic acid, were measured by high-performance liquid chromatography (HPLC) (UltiMate 3000; Thermo Scientific, Waltham, MA, USA) equipped with UV detector [6]. The analytes were separated by CAPECELL PAK MG S5 C_18_ (250 mm × 4.6 mm, 5 μm) held at 40 °C. The eluent consisted of solvent A (water containing 0.1% phosphoric acid) and solvent B (methanol) with isocratic elution of 97.5:2.5 (*v*/*v*) for 30 min. The flow rate was 0.7 mL/min. The wavelength of the UV detector was 210 nm. Samples were filtered through 0.22 μm microporous membrane for determination. According to the quantification of organic acids, calibration curves for each compound were constructed using pure standards at different concentrations. The calibration curves on 13 organic acids are shown in Appendix A.

### 2.9. DPPH Radical Scavenging Activity

This was carried out according to the Blois method with a slight modification [31]. Briefly, 0.02 mg/mL DPPH-ethanol radical solution was prepared, and then 2 mL of this solution was mixed with 2 mL sample. After 30 min of light avoidance reaction, the absorbance was measured at 517 nm by UV-1780 spectrophotometer (Shimadzu, Kyoto, Japan). Appropriate solvent blanks were performed in each assay. All determinations were carried out in triplicate, on each occasion and for each separate concentration for the standard and samples. The DPPH radical scavenging rate was calculated as follows: (2)$$\mathrm{DPPH}\;\mathrm{radical}\;\mathrm{scavenging}\;\mathrm{rate}\;\left(\%\right)=\left[\frac{A_{0}-A_{1}}{A_{0}}\right]\times 100$$
where *A*_0_ is the absorbance of the DPPH solution, and *A*_1_ is the absorbance of the mixture of DPPH solution and samples.

### 2.10. Statistical Analysis

Data were analyzed using Origin 2021, IBM SPSS Statistics 20.0 and Excel 2010. The statistical analyses were performed using one-way analysis of variance (ANOVA) followed by Duncan’s and Least significant difference (LSD) tests. Results were expressed as the mean ± standard deviation of three independent data in the diagram. If the different level was *p <* 0.05, it was considered to be significant. The Pearson method was used for correlation analysis. The clustering method was inter-group linkage.

## 3. Results and Discussions

In this study, the flora natively existing in BW was involved during spontaneous fermentation, regarded as the age-old method of homemade vinegar production. Nevertheless, the fermentation takes longer to execute compared to modern processing methods [32]. Currently, it is observed that the interest in fermented food by traditional production methods is gradually returning [33].

Regrettably, only a few reports on vinegar obtained via the spontaneous fermentation methods were observed. Mimura et al. [21] reported that sugar cane vinegar obtained by the naturally fermented method showed a great antioxidant potential compared with red wine vinegar from the inoculation method. Similarly, Ubeda et al. [34] reported that persimmon fruit vinegar obtained by spontaneous fermentation also showed a higher antioxidant potential compared with inoculated fermented vinegar. Nevertheless, these differences were not statistically significant.

### 3.1. Spontaneous Fermentation of BW Vinegar

Spontaneous fermentation of BW was executed in the laboratory. The dynamics of fermentation process for BW vinegar were determined based on changes in total acid and total sugar. As important indicators, total acid and total sugar affect both microbial growth and metabolite accumulation during vinegar fermentation. As shown in Figure 1, total acid gradually increased from 4.03 to 25.17 mg/mL during the first 30 days of fermentation. However, the production of total acid started to decrease generally after this time. At the 40th and 55th days, the production of total acid slightly increased again. The decrease in total acid content was probably due to the oxidation of organic acids by the oxygen present in the fermentation matrix or the metabolic conversion by microorganisms into other metabolites [35]. Total acid fluctuated throughout fermentation process, which is consistent with the previous spontaneous fermentation of date vinegar [35]. In general, the spontaneous fermentation of BW vinegar by the simply natural fermented process is slightly slow and requires about 60 days for a complete fermentation. 

In the first 20 days, total sugar, as a carbon source consumed by the microorganism, significantly decreased from 109.76 to 2.80 mg/mL, which may be owing to its utilization by microorganisms for cellular growth and bioconversion into organic acids [36]. During fermentation, the consumption of reducing sugar was very similar to total sugar (Figure 2). The content of reducing sugar was 57.81 mg/mL on the day 0 and 1.24 mg/mL on the 60th day, respectively, which was reduced by 97.86% (Appendix A).

### 3.2. Total Anthocyanins, Flavonoid and Polyphenol Contents

The presence of anthocyanins as the pigment is characteristic of BW varieties and eco-geographical environments [37]. BW varieties from Qinghai province contain considerable amounts of petunidin [38]. The dynamic of total anthocyanins during spontaneous fermentation were appeared in Figure 2. In general, total anthocyanins contents of samples ranged from 0.71 to 0.67 mg/mL with the reductions of 5.63% (Appendix A) also being accompanied by a change in coloration from purplish red to light red to red. The reasons for the reduction were mainly attributed to the oxidative degradation of anthocyanins itself occurred in fermentation [39,40]. Additionally, it could potentially result from the presence of different yeast strains that have different capacities to adsorb anthocyanins involved in spontaneous fermentation [41,42]. Interestingly, the content of anthocyanin decreased significantly from the 25th to the 30th day and then increased, especially after the 40th day in the fermentation. For this phenomenon, as far as we can guess, it may be due to the adsorption and release effect on anthocyanins of microorganisms (i.e., yeast) [41,42], the exact cause of which needs to be further investigated. Totally, BW vinegar produced by spontaneous fermentation had higher total anthocyanin content (0.67 mg/mL) than previously reported in red wine (202.33 mg/L) [43].

Unlike anthocyanins, total flavonoid and total polyphenol contents of BW vinegar increased notably after spontaneous fermentation with increases of 311.34% and 42.91%, respectively (Figure 2 and Appendix A). It has been reported that the total flavonoid and total polyphenol contents were increased during fruit juice fermentation through *Lactobacillus plantarum* MCC 2974 [44]. In this study, there was a dramatic fluctuation in total flavonoid and total polyphenol during spontaneous fermentation. Total flavonoid and total polyphenol increased significantly from the 0th to the 15th day and rapidly decreased from the 20th to 25th days to the minimum (2.08 and 2.30 mg/mL, respectively). In the end, the total flavonoid and total polyphenol contents were 3.99 and 3.93 mg/mL, respectively. This result can be attributed to the depolymerization of macromolecular polyphenol or the conversion of individual polyphenol compounds conducted by LAB strains [45,46]. Additionally, it has been suggested that the increase in the free forms of polyphenolic components and the production of other metabolites in the spontaneous fermentation may be responsible for the improvement of the antioxidant capacity [12]. Previous studies have indicated that fermentation could commonly increase the content of polyphenol. For instance, Liu et al. [24] found that total polyphenol contents increased during the fermentation of wolfberry juice with *Bacillus velezensis*, *Bacillus licheniformis* and *Lactobacillus reuteri* mixed with *Lactobacillus rhamnosus* and *Lactobacillus plantarum*. In addition, enzymes and carboxylic acids may have destroyed the cellular structure to release polyphenol compounds during fermentation, which led to increasing total polyphenol contents [24].

### 3.3. Total Protein, SOD and Amylase

Some proteins resulting from their digestion have been shown to be able to decrease oxidant effects [47]. The concentration of total protein increased significantly from 0.51 mg/mL to 1.37 mg/mL (the maximum) in the first 10 days, then subsequently decreased (Figure 2 and Appendix A). After whole spontaneous fermentation, total protein decreased by 27.45% in BW vinegar. According to the increase in total protein at the early stage of spontaneous fermentation, a similar result from the fermentation of wolfberry with lactobacillus and two bacillus species was also obtained by Liu et al. [24], where they discovered that fermentation increased the protein content in fermented wolfberry by 31.18% with a short fermentation duration. The decrease in total protein was mainly due to some bacterial strains producing various enzymes, such as proteases and peptidases, during fermentation [48]. Proteases and peptidases can break down macromolecular proteins with hydrolysis and convert them into smaller peptides and amino acids as a nitrogen source for microbial survival [49,50]. Therefore, it is possible that total protein contents is decreased in the BW vinegar due to the action of enzymes.

During the first 10 days of fermentation, SOD activity increased from 514.65 U/mL to 1725.46 U/mL (the maximum), and it reduced to 754.35 U/mL at the end (Figure 2 and Appendix A). Generally, SOD activity was enhanced by 46.6% after 60 days spontaneous fermentation. Apparently, SOD activity of both the maximum and final product was higher than apple juice fermentation using commercial strains [51]. SOD plays an important role in the improvement of antioxidant activity that relates to the presence of yeast and lactobacillus during fermentation [51]. Hence, BW vinegar via spontaneous fermentation showed significant antioxidant capacity, which demonstrated its potential as a health-promoting drink for dietary supplement and should be further developed.

Changes in amylase activity during spontaneous fermentation in BW vinegar showed a slow upward trend, rising from an initial 157.38 U/L to 196.46 U/L in the end, which increased by 24.83% (Figure 2 and Appendix A). The reason may be due to insufficient carbon sources in the fermentation system to promote amylase production by microorganisms.

### 3.4. Organic Acid

The composition and quantity of organic acids are important contributors to the flavour of fruit vinegar. In this study, a total of 11 organic acids, including lactic acid, acetic acid, gallic acid, sorbic acid, ascorbic acid, succinic acid, oxalic acid, malic acid, citric acid, tartaric acid and GABA were determined over spontaneous fermentation processing (Figure 3). Nevertheless, fumaric acid and shikimic acid were not detected (Appendix A). Generally, total contents of organic acids showed an increase followed by a steady state (Figure 3), while the decrease at the 45th day was probably due to the oxidation of organic acids or the metabolic conversion by microorganisms into other metabolites [35].

Initially, only eight organic acids were detected in the samples, such as gallic acid, GABA, oxalic acid, citric acid, succinic acid, malic acid, sorbic acid and tartaric acid. Especially, six organic acids, including oxalic acid, succinic acid, malic acid, tartaric acid, GABA and sorbic acid, accounted for the main part. Subsequently, oxalic acid, succinic acid, malic acid and tartaric acid significantly decreased (*p* < 0.05). In addition, succinic acid and tartaric acid were only present in the raw juice but absent after the 5th day and 30th day, respectively.

Lactic acid was not detected at the zeroth day, but it increased significantly throughout the rest of fermentation from 10.279 to 24.314 mg/mL and was the most abundant organic acids in the BW vinegar, consistent with the studies of Ren et al. [6]. The malic acid content decreased significantly from 0.460 mg/mL to 0.018 mg/mL during the early stage of fermentation. Overall, the production of lactic acid was closely related to the decrease in malic acid, indicating that malolactic fermentation occurred during the fermentation of BW vinegar [46].

Citric acid is an intermediate metabolite of the tricarboxylic acid cycle [52]. The transformation of intermediates into acetic acid or lactic acid by lactobacillus is likely responsible for the observed significant increase in acetic acid content during fermentation [53].

Ascorbic acid content was always under 0.810 mg/mL until the 40th day, and it was accumulated to reach the maximum of 5.170 mg/mL at the 55th day. Acetic acid was also a metabolite but was consumed up from the 40th to the 55th day. Additionally, the contents of gallic acid and GABA increased during fermentation, whereas the sorbic acid content was maintained between 0.372 and 0.437 mg/mL. Gallic acid, as a phenolic acid, increased during fermentation. The reason for this may be that the acidic environment created by *Lactobacillus* fermentation reduced the oxidation of polyphenols. The hydrolytic enzymes of *Lactobacillus* hydrolyze complex polyphenol compounds into simpler forms, i.e., deglycosylated polyphenols, and release bound polyphenol compounds from plant cell walls [54]. After the spontaneous fermentation process, the contents of most organic acids in BW vinegar were significantly higher than that in BW juice (*p* < 0.05).

### 3.5. PCA Analysis

Principal component analysis (PCA) was applied to establish a model and indicated the change tendency from multivariate data of metabolites during fermentation. The angles between the arrows indicated the degree to the similarity of BW fermentation explained by the metabolites and the length of the arrows indicated the extent to which the metabolites were correlated. Principal component 1 (PC1) and PC2 explained 43.9% and 20.9% of the variance, respectively (Figure 4). As can be seen in Figure 4, BW vinegar samples from different fermentation stages could be approximately separated into three different categories, namely, 0, 5–25 and 30–60 days, and the parallel samples representing similar characteristic from the same fermentation stages could be clustered well through quality assessments. The PCA also indicated that the maximum change in metabolites occurred from the zeroth day to the fifth day. According to Figure 4, sugars and organic acids (tartaric acid, succinic acid and oxalic acid) contributed the most to sample segregation for the zeroth day. Total anthocyanins, total protein, SOD and organic acids (acetic acid and citric acid) had greater impact on sample gathering for the 5th to 25th days, in which citric acid mainly contribute to the 5th day of fermentation and total protein and SOD to the 10th day. Additionally, total polyphenol, amylase, total acid, total flavonoid and organic acids (sorbic acid, gallic acid, ascorbic acid and lactic acid) were the main factors of sample gathering for the 30th to 60th days, among which total polyphenol, gallic acid and ascorbic acid mainly contribute to the 50th–60th days of fermentation. In general, samples from the 30th to 60th days gathered together, showing the high similarity of metabolites and illustrating the quality and stability of BW vinegar products through spontaneous fermentation.

### 3.6. Antioxidant Capacity

It is well known that antioxidant attributes of fruits are associated with the presence of natural antioxidants such as polyphenols, flavonoids, anthocyanins, ascorbic acid, SOD and so on [6,45,52]. The spontaneous fermentation of BW vinegar reinforced the DPPH free radical scavenging rate. Specifically, the DPPH free radical scavenging ability greatly increased from the 0th to 15th day and tended to stabilize from the 20th day, showing a good DPPH free radical scavenging ability (Figure 2). Until the 60th day, DPPH free radical scavenging ability reached 63.93%. The strength of antioxidant activity depends on the molecular weight of the polyphenols, flavonoids, organic acids and other related properties [6,24]. According to Figure 2, Figure 3 and Figure 4, it can be found that microbial growth was complicated and metabolic activities were fierce from the 0th to 25th day. This led to a significant difference in various metabolites. Therefore, the fermentation time should be prolonged over 25 days so that functional compounds, organic acids and antioxidant activity are available in abundance and quantity in the fermentation matrix.

### 3.7. Correlations among Metabolites and Antioxidant Capacity

Pearson correlations between metabolites and antioxidant activity were investigated and the results were presented in Figure 5. In general, the main functional components in BW vinegar, including GABA, lactic acid, total acid, total flavonoid, gallic acid and total polyphenol, were correlated with the values of the DPPH free radical scavenging ability (DPPH) (*p* < 0.05). Specifically, there were highly significant correlations (*p* < 0.001) between DPPH and GABA (*R*^2^ = 0.77), DPPH and lactic acid (*R*^2^ = 0.79), DPPH and total flavonoid (*R*^2^ = 0.76) and DPPH and total acid (*R*^2^ = 0.81). Additionally, the components of gallic acid (*p* < 0.01) and total polyphenols (*p* < 0.05) had correlations with DPPH (antioxidant test). As the main functional components in BW vinegar, GABA, lactic acid, total flavonoid, total acid, gallic acid and total polyphenol differently contributed to antioxidant capacity. Dávalos et al. [55] reported that the antioxidant capacity and polyphenol compounds had a positive correlation in wine vinegars (*p* < 0.01), consistent with our finding. The reason is probably that the transformation of polyphenol compounds into other new metabolites with antioxidant capacity [56] and flavonoid glycosides were transformed into corresponding glycosides with strong free radical scavenging ability through specific bacterial glycosyl hydrolases during fermentation, so that the antioxidant properties were increased at the end of fermentation [57,58].

Although anthocyanins belong to polyphenol, the negative correlation between anthocyanins and DPPH (*R*^2^ = −0.44, *p* < 0.01) has been noticed, mainly owing to the degradation of anthocyanins. DPPH was significantly positively correlated with total acid and total flavonoid (*p* < 0.001), which indicated that antioxidant capacity mainly came from total flavonoid. The effect of DPPH free radical scavenging was also affected by organic acids, specificly, GABA, lactic acid and gallic acid (*p* < 0.01). The antioxidant activity of vinegars may be attributed to these present organic acids apart from acetic acid, including gallic acid, caffeic acid, syringic acid, citric acid, chlorogenic acid, tartaric acid and ferulic acid [32,59,60]. Short fatty acids, such as lactic acid and acetic acid, possess antimicrobial and nutritional benefits [58]. This study indicated that BW vinegar possessed abundant organic acids with different and complex compositions. These organic acids can exert some health benefits such as antimicrobial activities [32]. It was discovered that fruit vinegars include more complex compositions of organic acids than cereal vinegars.

Berry vinegars are popular worldwide for their good flavor profile and health benefits. Some studies have indicated that different health benefits of fruit vinegars [14]. Due to the difference of polyphenol composition and contents in BW vinegar, further studies are required to explore the bioavailability and health benefits of BW vinegar in vivo. For that reason, the antioxidant capacity of BW vinegar was comprehensively evaluated through changes in antioxidants, organic acids, free radical scavenging capacity and SOD activity. An interesting result found in this research was that the contents of total polyphenol and total flavonoids were improved, but SOD activity was not the main antioxidant in fermentation (Figure 2 and Figure 5). In addition, although the content of antioxidants and organic acids have changed to a greater or lesser degree, the scavenging ability on DPPH free radicals of BW vinegar was enhanced after spontaneous fermentation (Refer to Figure 2 and Appendix A). This analysis is consistent with pomegranate juice [58]. Simultaneously, notwithstanding values were correlated with each other, further studies are still needed to evaluate the specific compounds that contribute to each value in BW vinegars, as most of methods were based on the same reaction mechanism.

## 4. Conclusions

The composition of BW vinegar differed in terms of total acid, sugar, anthocyanin, polyphenol, flavonoid, protein, SOD, amylase, organic acids and antioxidant capacity during spontaneous fermentation. Significant differences were observed in the analysis of antioxidant capacity among BW vinegars collected at each fermentation point. Functional compounds including total flavonoid, total polyphenol and three organic acids (GABA, lactic acid and gallic acid) played the main role in antioxidation, whereas SOD and ascorbic acid (vitamin C), known as antioxidants, did not correlate with DPPH free radical scavenging ability, reaching the contents of 754.35 U/mL and 3.39 mg/mL, respectively. Furthermore, acetic acid content was consistently maintained at a lower level compared with other fruit vinegars, which indicated that the fermentation parameters still need to be optimized. Through analyzing changes in metabolites and antioxidant capacity in vitro during the spontaneous fermentation of BW vinegar, it is confirmed that the contents of functional compounds and organic acids have been improved. In the future, based on these research results, the screening and identification of dominant microorganisms in the spontaneous fermentation of BW vinegar should be performed.

## Figures and Tables

**Figure 1 foods-11-01344-f001:**
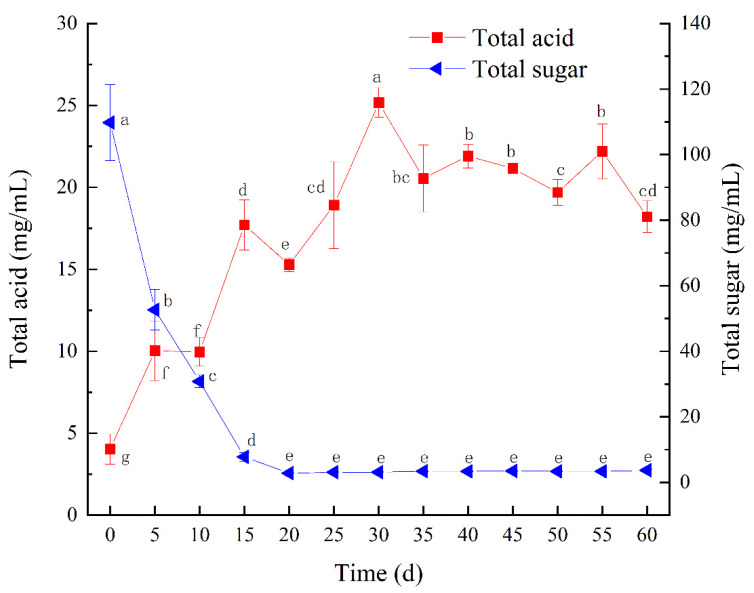
The dynamic of total acid and total sugar in BW vinegar fermentation. Different letters are significantly different for the same index (*p* < 0.05).

**Figure 2 foods-11-01344-f002:**
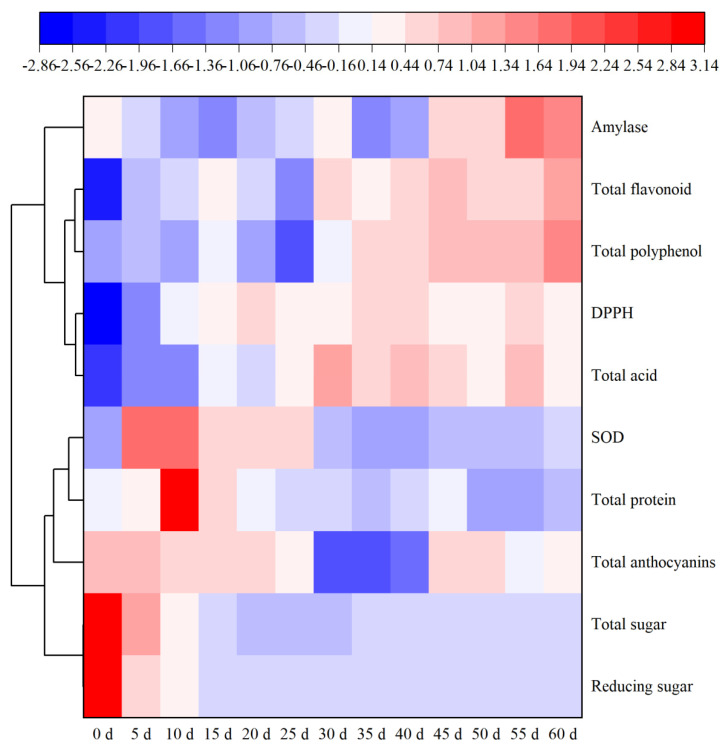
Heatmap on the dynamic of metabolites and DPPH free radical scavenging ability (DPPH) during BW vinegar fermentation. Blue represents relatively low content levels, and red represents relatively high content levels.

**Figure 3 foods-11-01344-f003:**
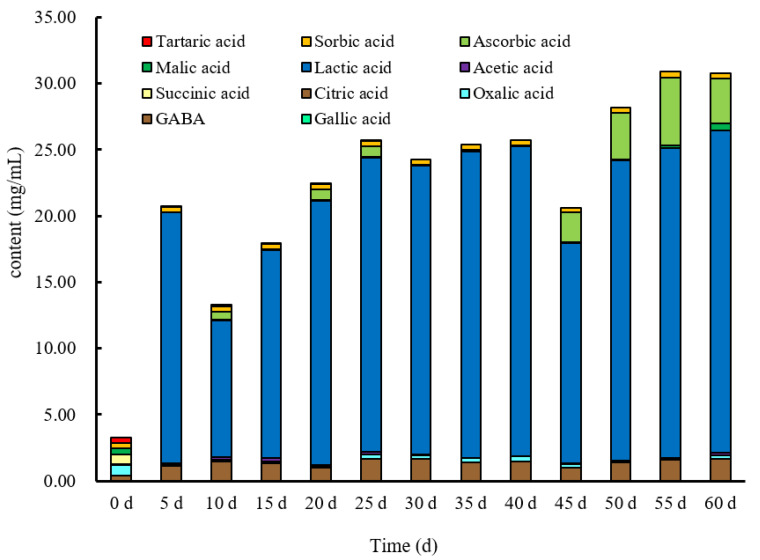
The profile of organic acids contents in BW vinegar fermentation.

**Figure 4 foods-11-01344-f004:**
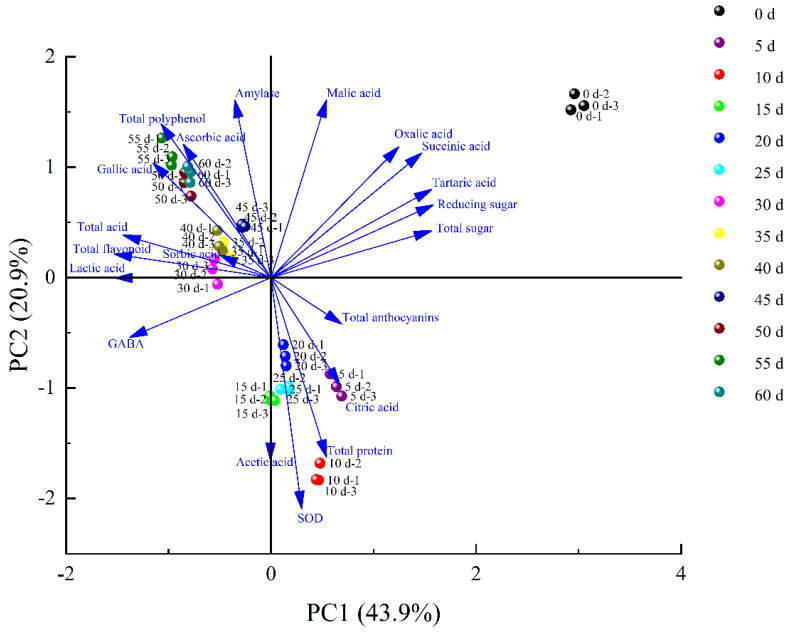
PCA analysis of metabolites during fermentation with the optimized model.

**Figure 5 foods-11-01344-f005:**
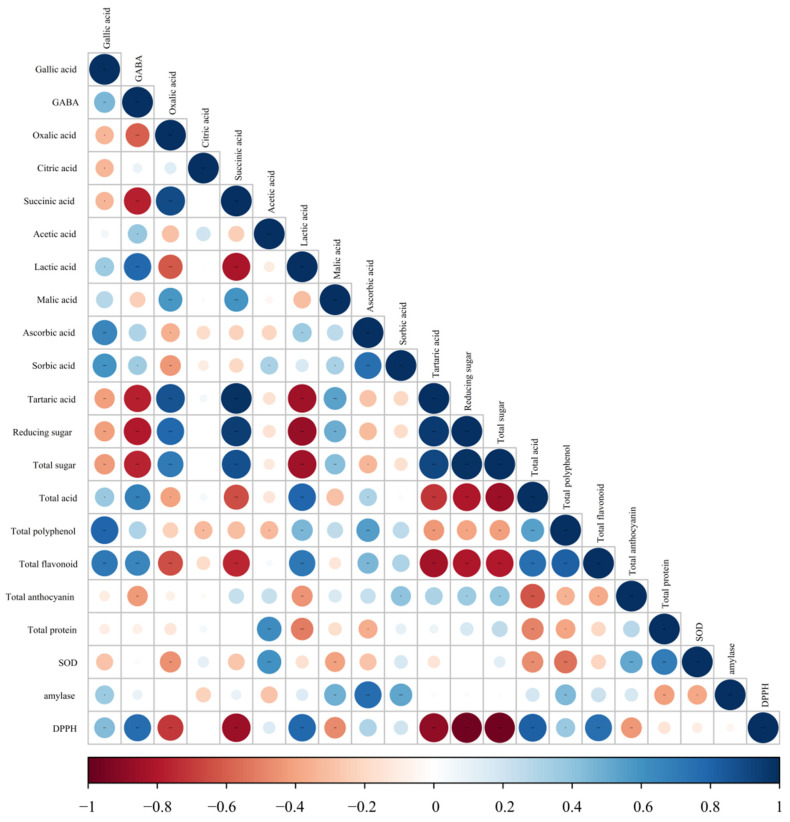
The bubble diagram of correlations among metabolites and antioxidant capacity during BW vinegar fermentation. Blue represents a positive correlation, and red represents a negative correlation. * 0.01 < *p* ≤ 0.05, ** *p* ≤ 0.01, *** *p* ≤ 0.001.

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
