# Peer review of "Dynamics of Physicochemical Properties, Functional Compounds and Antioxidant Capacity during Spontaneous Fermentation of Lycium ruthenicum Murr. (Qinghai–Tibet Plateau) Natural Vinegar"

_foods, 2022, doi:10.3390/foods11091344_

Round 1
Reviewer 1 Report
The introduction contains enough information, but please check if it is necessary to repeat the content of the sentence "In fact, supplementing
dietary natural antioxidants have become an imperative strategy for curing diet-related diseases, reducing oxidative stress, preventing age-related diseases, and maintaining homeostasis [16]" twice in the same paragraph (similar sentence is above this one).
Figure 3 is pretty hard to read, maybe try some other form of graphical design?
The paper was written correctly, with a detailed description of the goal, methods and obtained results. The subject is very interesting and has potential for further research (different yeasts for example).
However, a plagiarism detection software found a total of 43% similarities with other articles. Please check and refine the text.

Reviewer 2 Report
Dear Editor and Authors,
I send you my review about the article “Dynamics of physicochemical properties, functional compounds and antioxidant capacity during spontaneous fermentation of Lycium Ruthenicum Murr. (Qinghai-Tibet Plateau) natural vinegar”.
The main scope of the research, as reported in the aim was to evaluate the dynamics of physicochemical properties, functional compounds and potential health effects of the vinegar obtained from Lycium Ruthenicum.
In my opinion, the article, although it result sufficiently well written it need a little revision of the English language.
Moreover the introduction result to much long and some aspect may have been over-emphasized. For these reasons I suggest to the authors to delete the non-essential concepts and phrases.
In particular the below sentence could result speculative and they should be removed.
“Black wolfberry (BW, Lycium ruthenicum Murr.) is a nutritional and dark purple berry of Solanaceae, widely distributing in the Qinghai-Tibet Plateau. BW is a traditional and medicinal herb since it’s rich in nutritional and functional components (soluble sugars, organic acids, minerals, vitamins, anthocyanins, polyphenols, flavonoids, polysaccha- rides, and carotenoids) [1-3]. It is mainly consumed fresh or dried. As a therapy plant, its ripe berries are used to treat abnormal eye disease, menopause, menstruation and hypertension, which are highlighted in the classical “Compendium of Materia Medica” and “The Four Medical Tantras”. Therefore, BW products (traditional medicines, nutri- ents and functional food or beverage) emerge in enormous numbers due to many health- related aspects resulting from the consumption of BW and products prepared by BW [1]. Although the beneficial effects presented by BW were noticeable, the importance as a source of antioxidant compounds should be also emphasized.”
Furthermore, the sentences from “the quality of natural vinegar…” to “…SOD and organic acids” should be supported by references.
The paragraph of materials and methods should be improved adding some information.
Indeed, in the materials and methods paragraph should be reported the number of the fermentation trials made and the sampling procedures.
Furthermore, in my opinion, the difference among the number of the experimental trials and the number of replicate of sampling per each trial should be better highlighted.
Furthermore, the method for the determination of the total anthocyanins content, for the determination of organic acids should be supported by references.
Moreover, always in materials and methods the method used to determine the statistichals significance of differences should be reported (for example the Authors should report if they used the one wey analysis of variance or the repeated measures analysis of variance ecc.).
The results is well presented and they are complete discussed, also in comparison to the data reported in the literature.
However, to facilitate the reading of the graph shown in the figure 1 by the readers it should be highlighted, in the graph or in the legend, the difference among the means values (for examples using different letters).
Furthermore for a better understanding of the data shown in the figures 3 I suggest to the authors to replace the graph with a table.
Finally, the conclusions resulted adequate to the data showed and to the aim of the research.
Best regards
Reviewer 3 Report
Comments to authors
Journal: Foods
Title: ‘Dynamics of physicochemical properties, functional compounds and antioxidant capacity during spontaneous fermentation of Lycium Ruthenicum Murr. (Qinghai-Tibet Plateau) natural vinegar’
The current article focused on the physicochemical properties of functional compounds as well as potential their health effects. The research is innovative and interesting, but still, I have fewer minor questions as mentioned below.
Minor comments:
- The article needs to be revised thoroughly by English experts as it contains many syntax and grammatical errors.
- For example, on page # 2 ‘And no research has been reported on the interaction between bioactive ingredients and characteristics physicochemical properties of natural vinegar made from BW during spontaneous fermentation’. This sentence needs to be rewritten.
- Page # 3 heading 2.2 ‘All berries were at the optimum point of
maturity and sound quality without disease’’ Rewrite this sentence. - Authors should provide a reference for vinegar preparation.
- Page # 3 heading 2.2 ‘After filtration, samples were centrifuged (3000 r/min, 10 min) and stored in a -80 ℃ fridge for testing’’ revise this sentence.
- Page # 3 authors should write the symmetrical equation numbers to the throughout used equations in the manuscript.
- Page # 4 heading 2.5, Sentence needs to revise ‘Finally, the absorbance values were
measured at 510 nm. Using rutin as a standard, 0.10 mg/mL rutin standard reserve fluid
was prepared with 95 % ethanol and a series of rutin standard solutions were diluted with
70% ethanol for testing’. - Rewrite it Page # 4 heading 2.8 ‘CAPECELL PAK MG S5 C18 (250 mm × 4.6 mm, 5 μm) was used as chromatographic column with temperature of 40 ℃’.
- Under results and discussion page # 5 ‘Nevertheless, differentiating fruit flora that took part in the fermentation may have influenced the differentiation of obtained results. This sentence should be rewrite
- Page#7, heading 3.2, rewrite this sentence ‘On the one hand, it is well
known that oxidative degradation of anthocyanins itself occurred in fermentation. On the
other hand, this could be also resulted from the presence of different yeast strains involved
in spontaneous fermentation’.
Main comments:
- The overall article needs to be revised by a native English speaker.
- The total acid values are continuously fluctuating from day first to day 60, authors should explain the valid technical reason behind this phenomenon.
- introduction of the manuscript should be improved and well-structured there are few grammatical errors throughout the manuscript.
- Results should be well defined and the discussion part is not closely related to the results, and the focus of the discussion is not focused. So, improve the discussion part according to the current and previous studies.
- Why total anthocynin has been flucatuating during day 25 to day 40.
- During the fermentation there is melodramatic variations are showing in total flavonoid, total polyphenol and amylase authors should explain the valid reason in the manuscript under the heaing of results and discussion.
- Authors should be add more comparative discussion with previous studies under the heading of Organic acid. I think it will be more easy to understand the concept for readers.
- Organic acid is decresead after 45th day what is the reason behind this strange behaviour.
Reviewer 4 Report
The paper describes fermentation of black wolfberry and the influence of this process on chemical composition of the product (i terms of classes of compounds). some improvements are acknowledged.
1./ Paragraph devoted to PCA analysis should be carefully rewritten. As far as I understand three samples were studies by PCA analysis regarding the changes in all (?) metabolites versus time. If "0 d" represent starting points the changes seem to be big in a different manner. Some more throughful dissccussion has to be done. Additionally there is a fourth grouping of results for 3rd day, which do not fit to general trend. - is it outliner? Is there any reason for that?
Additionally, it would be desirable to combine the two plots shown in Figure 4 as a plot C showing interations for better illustration of the analysis.
2./ I am not sure that fermentation process (paragraph 3.2.) should be so detaily described
3./ Other small comments are:
- some reference or explanation should be provided to “Compendium of Materia Medica” and “The Four Medical Tantras” (source, age of publication);
- sentence "In recent years, researchers have pro-duced natural vinegar from fruit sources such as grape, lemon [5]" shouyld be changes since reserachers rather were involved in studies on vinegars produced from many fruits in their native countries;
- table 1 should be discarded or moved to supplementary material;
- Figure 1: inside the graph should be "sugar" not "suger";
- when describing anthocyanin changes Authors did not mention about important changes in coloration (are there any?);
- Figure 3 would be better to follow if it would be coloured;
- Conclusions: sentence about tnhe fact that fermented black wolfberry vinegar is a potential source of antioxidants is trivial.
Round 2
Reviewer 1 Report
Dear researchers,
thank you for your responses and revision. I recommend this paper to be published in Foods.
Best regards